# Outcome Prediction in Critically-Ill Patients with Venous Thromboembolism and/or Cancer Using Machine Learning Algorithms: External Validation and Comparison with Scoring Systems

**DOI:** 10.3390/ijms23137132

**Published:** 2022-06-27

**Authors:** Vasiliki Danilatou, Stylianos Nikolakakis, Despoina Antonakaki, Christos Tzagkarakis, Dimitrios Mavroidis, Theodoros Kostoulas, Sotirios Ioannidis

**Affiliations:** 1Sphynx Technology Solutions, 6300 Zug, Switzerland; 2School of Medicine, European University of Cyprus, 2404 Nicosia, Cyprus; 3School of Electrical and Computer Engineering, Technical University of Crete, 73100 Chania, Greece; snikolakakis@tuc.gr (S.N.); sotiris@ece.tuc.gr (S.I.); 4Institute of Computer Science (ICS)-Foundation for Research and Technology-Hellas (FORTH), 70013 Heraklion, Greece; despoina@ics.forth.gr (D.A.); ctzagkarakis@gmail.com (C.T.); dmavroid@gmail.com (D.M.); 5Department of Information and Communication Systems Engineering, School of Engineering, University of the Aegean, 83200 Samos, Greece; theodoros.kostoulas@aegean.gr

**Keywords:** venous thromboembolism, cancer, mortality, ICU, interpretable machine learning

## Abstract

Intensive care unit (ICU) patients with venous thromboembolism (VTE) and/or cancer suffer from high mortality rates. Mortality prediction in the ICU has been a major medical challenge for which several scoring systems exist but lack in specificity. This study focuses on two target groups, namely patients with thrombosis or cancer. The main goal is to develop and validate interpretable machine learning (ML) models to predict early and late mortality, while exploiting all available data stored in the medical record. To this end, retrospective data from two freely accessible databases, MIMIC-III and eICU, were used. Well-established ML algorithms were implemented utilizing automated and purposely built ML frameworks for addressing class imbalance. Prediction of early mortality showed excellent performance in both disease categories, in terms of the area under the receiver operating characteristic curve (AUC–ROC): VTE-MIMIC-III 0.93, eICU 0.87, cancer-MIMIC-III 0.94. On the other hand, late mortality prediction showed lower performance, i.e., AUC–ROC: VTE 0.82, cancer 0.74–0.88. The predictive model of early mortality developed from 1651 VTE patients (MIMIC-III) ended up with a signature of 35 features and was externally validated in 2659 patients from the eICU dataset. Our model outperformed traditional scoring systems in predicting early as well as late mortality. Novel biomarkers, such as red cell distribution width, were identified.

## 1. Introduction

Venous thromboembolism (VTE) and cancer are major causes of death worldwide [1] and their prevalence is continuously rising due to the doubling of life expectancy [2], the tripling of the world population over the last 70 years, changes in lifestyle, the increased prevalence of chronic diseases [3], and the COVID-19 pandemic [4]. VTE can present with clots in the veins, most frequently as deep vein thrombosis (DVT) and pulmonary embolism (PE). Despite the recent advances in treatment, fatality rates have increased in the last decade, with notable racial and geographical disparities [5]. Patients with VTE or cancer occasionally need advanced support and suffer from significant morbidity, prolonged intensive care unit (ICU) stay, and high mortality rates, not only early during hospitalization but even after several months.

Mortality prediction in ICU patients has been a major challenge in the area of medical informatics [6] and has long been used as a quality indicator. Mortality is a major end point in epidemiological and interventional studies in the ICUs [7]. Published studies so far mainly focus on the prediction of in-hospital or early mortality, irrespective of the primary diagnosis and related comorbidities [8,9,10,11,12], whereas studies focusing on late mortality prediction are quite rare [13,14]. It is generally accepted that the initial diagnosis of a patient and the evaluation of the reasons for ICU admission could significantly affect the overall survival; thus, it would be interesting to study individually the different disease outcomes and identify the specific disease-related clinical features with prognostic significance. Even more importantly, it is necessary to predict post-discharge mortality, which is a very challenging task, since patients admitted to the ICU usually suffer from a high comorbidity burden [15]. Prompt identification of factors associated with late mortality could help physicians to re-orientate medical clinical practices (e.g., extended anticoagulation), identify modifiable factors, and reasonably allocate health resources in the ICUs, which are extremely restricted, especially during the COVID-19 pandemic period.

On the other hand, there is a growing availability of automated ICU patient surveillance systems and healthcare big data that remains unexploited, whereas they could provide the opportunity to examine data-driven research solutions using modern machine learning (ML) methods. The wealth of available ICU data combined with the use of ML-based approaches have impacted medical predictive analytics and clinical decision support systems, since ML algorithms can learn from complex data patterns and identify associations that could help in the improvement of patient care and survival, as well as lowering hospitalization costs [16].

This article is organized as follows. Section 2 reviews previous work on ICU mortality prediction. Specifically, traditional clinical scores considering the standard clinical practice based on a limited number of features, collected during admission to the ICU, are compared with recent approaches using state-of-the-art ML algorithms and routinely collected data from medical records. Section 3 presents the motivation and the aim of our work, which is early and late mortality ML-based prediction focusing on patients with thrombosis or cancer. Section 4 describes the datasets used in our study, the selection of cohorts, the time-dependent features used to build the prognostic model, the data manipulation, and the two adopted ML frameworks. Section 5 and Section 6 provide an extensive discussion of the results on early and late prediction in ICU patients with thrombosis or cancer, the main predictive features, along with their interpretation and the external validation of the model. Finally, Section 7 summarizes the main outcomes and highlights the fundamental contributions of the current study.

## 2. Related Work

Since the early 1980s, several clinical scores have been introduced into clinical practice to predict in-hospital ICU mortality. The Simplified Acute Physiology Score (SAPS) [17], Acute Physiology and Chronic Health Evaluation (APACHE) [18], and Sequential Organ Failure Assessment (SOFA) [19] score are considered as validated tools in predicting ICU mortality [20]. Nevertheless, these scores have several limitations. They have been developed based on different target populations with heterogeneous inclusion and exclusion criteria; thus, during the validation process, they provide modest performance. They are based on multivariate statistical methods, such as logistic regression models, disregarding the non-linear relationships that exist between variables in real-life medical data. Since the scores are computed on health data collected during the first 24 h of ICU admission or instant-based measurements (e.g., the worst or average value), they do not consider time-based measurements, which could contain important information about clinical deterioration [21]. It is worth noting that inter-rater agreement is low between the various clinicians, mostly affected by personal experience, and thus a potential bias in the scores’ interpretation exists [22]. Finally, these clinical prediction scores have poor generalization and inadequate model calibration, especially in the high-risk patients, which is the type of patients that we study [23].

Other thrombosis-specific clinical scores, such as the Pulmonary Embolism Severity Index (PESI) [24], have been mostly used to identify low-risk patients that could benefit from outpatient treatment or early discharge, although they have also been validated to assess the probability of 30- and 90-day mortality post-PE [25,26,27,28]. Overall, it is unclear if clinicians routinely use these risk stratification tools, since only a small proportion of low-risk patients based on the PESI score are discharged earlier [24]. Thus, the clinical usefulness of these risk stratification scores needs to be proven. A recent meta-analysis combining all available risk stratification tools for PE showed that although most of them have high sensitivity, the low specificity in the range of 50% discourages clinicians from universal acceptance and employment in everyday clinical practice [29].

There is increasing interest in the literature in using modern ML algorithms, such as random forests (RF) and gradient boosting machines, since it has been shown that they can predict more accurately clinical outcomes (e.g., mortality) in comparison with classical logistic regression [30]. Some of the recent works on mortality prediction in ICU patients show the superiority of the ML-based predictive models against the traditional clinical scores [10,31,32]. These studies are not focused on a specific diagnostic group of patients and only use a limited feature set [8,11,12,33].

Prediction of mortality in critically ill patients with thrombosis or cancer using ML algorithms has not been extensively studied so far. To the best of our knowledge, the only study in the literature that investigates ICU patients with thrombosis is [32]. The study is based on a relatively small number of patients, and the authors used only admission data, developing two models for 30-day mortality prediction based on logistic regression, and least absolute shrinkage and selection operator (Lasso) for feature selection. Regarding patients with cancer, there is a paucity of studies predicting mortality in the ICU. In [34], the authors provide a systematic review on ML-based early mortality prediction in cancer patients (but not critically ill patients). The average area under the receiver operating characteristic curve (AUC–ROC) is 0.72–0.92 and differs between the various cancer types. The absence of studies in this field can be partly explained by the fact that the admission of cancer patients to the ICU is considered frequently unavailing, and physicians sometimes discourage their patients and families from proceeding with aggressive treatment. Nevertheless, it is highly important to recognize the patients who would benefit from intensive care treatment. Traditional clinical scores, such as APACHE, fail to predict accurately individual outcomes in these patients [35,36], and thus there is a need for more sophisticated models.

## 3. Motivation of the Current Work

Motivated by the lack of studies in the two particular diagnostic groups of patients mentioned above, as well as by the limitations of the traditional clinical scores, we aim to develop and validate prognostic models for early and late mortality using state-of-the-art ML algorithms, while exploiting almost all available data in the electronic health record. Since healthcare datasets are unstructured and heterogeneous, manual features’ engineering and extraction is time-consuming; thus, combined bioinformatician and clinician expertise is needed. This can be attenuated by adopting ML models trained on multidimensional data [33]. Our initial hypothesis was that the utilization of all the information stored in the electronic health record, i.e., demographics, laboratory tests, medications, diagnosis, and relevant procedures, along with their detailed timeline information, could help a clinician to dramatically improve the decision-making process. We were also interested in identifying the most informative features in a medical record, and thus we performed feature discriminative analysis during the analysis of the various groups of features.

Towards this direction, two different open-access high-dimensional multicenter retrospective datasets were used (MIMIC-III and eICU databases). The derived models were compared with traditional clinical scores, externally validated to prove generalizability, and finally interpreted by identifying clinically meaningful predictive features. Additionally, we introduced a custom-based ML approach combined with an oversampling method to address the dataset imbalance. We recognized some interesting biomarkers, which, although readily available, are currently disregarded during clinical practice. We strongly believe that our work will be of added value in refining conventional clinical scores and rediscovering easily measurable and low-cost biomarkers.

## 4. Materials and Methods

### 4.1. Ethics Statement

This is a retrospective study based on two freely accessible databases, MIMIC-III [37] and eICU [38], both created in accordance with the Health Insurance Portability and Accountability Act (HIPAA) standards, where all investigators with data access were approved by PhysioNet [39]. Patient data were de-identified. MIMIC-III patient data are date-shifted and age over 89 years old was set to 300 years old, whereas in eICU, older patients are referred to as >89 years old. All pre-processing and data analysis were performed under Physionet regulations.

### 4.2. Dataset Description and Cohort Selection

#### 4.2.1. MIMIC-III Database

The Medical Information Mart for Intensive Care Database (MIMIC-III, version 1.4) [37] is a single institutional ICU database from Beth Israel Deaconess Medical Center, comprising health-related data from 38,597 adult patients and 49,785 admissions in the ICU between 2001 and 2012. Two groups of patients were studied, namely patients with (i) thrombosis and (ii) cancer. Initially, we selected all patients aged >15 years old hospitalized in the ICU with a primary diagnosis of thrombosis based on 35 distinct ICD-9 codes. We excluded age <15 years (n=3), pregnancy and puerperium complications (n=40), and patients with a “do not resuscitate” (DNR) code (n=169). Overall, 2468 patients (6.4% of total patients in MIMIC-III) were selected. As a second group, we utilized all patients aged >15 years old hospitalized in the ICU with a primary diagnosis of solid cancer or hematological malignancy based on 101 different ICD-9 and ICD-10 codes. Overall, 5691 patients (14.74% of total patients in MIMIC-III) were selected. Patients with age <15 years (n=0), pregnancy and puerperium complications (n=15), and patients with DNR code (n=358) were excluded. After the exclusion, a total number of 5318 patients remained in the study.

#### 4.2.2. eICU Database

The eICU Collaborative Research Database (eICU, v2.0) [38] is a multi-center ICU database with over 200,000 admissions for almost 140,000 patients, admitted to more than 200 hospitals between 2014 and 2015 across the USA. We selected patients aged >15 years old with a primary diagnosis of VTE based on 18 different ICD-9 and ICD-10 codes. The same inclusion and exclusion criteria were used, as described above. Overall, 4385 VTE patients (3.15% of total patients in the eICU database) were identified. Detailed demographic and clinical characteristics of patients from both databases are shown in Table 1.

### 4.3. Feature Pre-Analysis Selection

To investigate potential novel discriminatory attributes, features that were extracted from the database and used to build the prognostic model were chosen based on the clinical experts’ opinion and, following a liberal approach, we tried to simulate a real-life scenario where the medical practitioner exploits all relevant clinico-laboratory information available on the electronic health record.

#### 4.3.1. MIMIC-III

A total amount of 1471 features including demographics, clinico-laboratory information, medications, and procedures were extracted. Apart from the features extracted directly from the database, we computed through SQL queries various meta-features (known as concepts), such as clinical severity scores and first-day labs available as scripts on GitHub. Unstructured notes written by clinicians in free text format were extracted as text entities using the Sequence Annotator for Biomedical Entities and Relations (SABER) [40], which is a deep learning tool for information extraction in the medical domain. A detailed description of these features’ and meta-features’ processing, as well as a natural language processing overview, can be found in [41].

#### 4.3.2. eICU

Out of 31 tables, we selected variables from the following tables: *patient* (demographics, admission, and discharge information), *diagnosis*, *admission_dx* (primary and other diagnoses), *physicalExam*, *vitalPeriodic* (vital signs), *apachePredVar*, *apacheApsVar* (clinical scores), *lab* (laboratory measurements), *admissiondrug*, *infusion*, *treatment* (drugs administered prior to and during ICU stay). The description of the selected features for each group is given in Appendix A
Table A1.

#### 4.3.3. Traditional Clinical Scores

The eICU database contains information from APACHE version IV and IVa and Acute Physiology Score (APS) version III scores, whereas the MIMIC-III database contains SOFA, SAPS, Outcome and Assessment Information Set (OASIS), Logistic Organ Dysfunction Score (LODS), Multiple Organ Dysfunction (MODS), APS III, as well as comorbidity scores such as Elixhauser. We compared the performance (AUC–ROC) of the derived model using the extended feature set with the available medical scores recorded in the databases.

### 4.4. Data Manipulation and Transformation

Age adjustment has been originally applied in both datasets to comply with privacy regulations. Older patients were all assigned as 90 years old, given that the risk of thrombosis is homogeneously high in those more than 85 years old.

JADBio automatically performs pre-processing of the data—that is, mean and mode imputation of missing data, constant feature removal (features that contain only one value for all the outputs in the dataset and therefore are meaningless are removed), and standardization of the feature range.

A significant challenge that had to be addressed manually during attribute selection was the redundancy of the features in both datasets. For example, several drugs are prescribed either with the trade name or the active compounds (e.g., enoxaparin or lovenox), whereas misspellings of the names are frequent (e.g., agatroban or argatoban instead of argatroban). The most important medication groups extracted were anticoagulants, antiplatelets, cardiovascular, antidiabetic, antilipidemics, thrombolytics, vasopressors, antibiotics, chemotherapeutics, and corticosteroids. Dosage and duration of treatment was not taken into account in the current experiments. Laboratory tests, such as complete blood count, kidney and liver function tests, acid base balance, clotting, and biochemical and enzyme tests, were extracted. For each of these features, the first and the average value from the whole ICU stay were selected. Time stamps were defined as the average values per 6 h, during the first 48 h of the ICU stay. Finally, vital signs were recorded as average values per 1 h, during the first 48 h in the ICU, first, and average values during the ICU stay.

### 4.5. Automated Machine Learning Framework: JADBio

The automated ML (AutoML) platform JADBio automatically tries and evaluates numerous ML pipelines, optimizing the pre-processing, feature selection, and modeling the steps and their hyper-parameters [42]. It employs the Bootstrap Bias Corrected Cross-Validation (BBC-CV) to provide an unbiased estimation of performance that adjusts (controls) for trying multiple ML pipelines [43,44]. Pre-processing, normalization, mode and mean imputation, constant feature removal, and feature selection are not applied to the complete dataset before cross-validation, thus avoiding overestimation and over-fitting. The classification algorithms used are linear, ridge, and Lasso regression, decision trees, random forests (RF), and support vector machines (SVMs) with Gaussian and polynomial kernels. For feature selection, JADBio uses the Lasso and Statistically Equivalent Signature (SES) [45] algorithms. JADBio applies best practices of ML to eliminate any over-fitting of the model and overestimation of its out-of-sample predictive performance, even for small sample sizes; a detailed evaluation of JADBio can be found in [43].

### 4.6. Custom Machine Learning Framework: XGBoost

The custom ML framework is depicted in Figure 1. A stratified train–test split of 80–20% is applied to the initial dataset, where the missing values are imputed by using mean imputation (for numerical features) and mode imputation (for categorical features). Then, the data preparation phase corresponds to the one-hot encoding of categorical features and correlation-based feature selection, where features with high Pearson correlation values are more linearly dependent, having almost the same effect on the dependent variable, and thus can be removed. Specifically, the pairwise Pearson correlation matrix is computed and the features with a correlation ratio higher than 0.9 are dropped. Bayesian optimization [46] is adopted during the hyper-parameter tuning and a stratified five-fold cross-validation strategy is followed. It is also important to notice that JADBio addresses imbalanced classes through stratified cross-validation and diversified class weights during SVM learning. Since the imbalanced ratio in the eICU dataset is 3457 vs. 267 class samples for classes 0 and 1, respectively, we aim at examining the class balancing effect in light of oversampling combined with a state-of-the-art ML classifier, where, in the current work, we investigate the XGBoost (eXtreme Gradient Boosting) classifier’s performance. Towards achieving a balanced ratio between the two classes, the Synthetic Minority Oversampling Technique (SMOTE) [47] is adopted. In each cross-validation fold, resampling is applied to the data, and as a final step, we use the hyper-parameters for which the cross-validation achieved the best performance as well as SMOTE on the total training set.

### 4.7. Machine Learning Models’ Performance Evaluation

The performance of ML classification tasks is typically assessed by various well-known metrics, such as sensitivity (or recall), accuracy, specificity, F1 score, and area under the receiver operating characteristic curve (AUC–ROC). The significance of the AUCs was measured using a significance level of a=0.05 [48]. The classification threshold was optimized for F1 score.

## 5. Results

In this section, we describe the predictive models of early and late mortality for patients with thrombosis or cancer, using an automated ML tool. Performance metrics, feature discriminative analysis, comparison with conventional scores, and validation of the model are reported in detail. In the last part of this section, the results of the custom ML framework that addresses the class imbalance problem are presented.

### 5.1. Automated Machine Learning

#### 5.1.1. Prediction of Early Mortality in ICU Patients with Thrombosis

As early or in-hospital mortality define the outcomes of patients at discharge from the hospital, two different datasets were used and two groups of patients with thrombosis were extracted: from the MIMIC-III database, 348 non-survivors vs. 1303 survivors, and from the eICU database, 267 non-survivors vs. 3457 survivors.

##### MIMIC-III Dataset

Prediction of early mortality in patients with thrombosis, using all clinico-laboratory features, achieved excellent performance (AUC–ROC = 0.93, CI 0.91–0.95), where CI denotes the confidence interval. The winning algorithm was random forest (RF), training 500 trees with the deviance splitting criterion and minimum leaf size = 3. Detailed information regarding metrics of performance and feature discriminative analysis can be found in [41]. As shown in Figure 2 and Appendix A
Table A3, our model significantly outperformed nine well-known traditional medical scores, such as SAPS [17] and SOFA [19]. Specifically, among the various scores, the best performance was achieved by SAPSII (AUC–ROC = 0.85, CI 0.81–0.89), and when a combination of all available scores was used, the achieved performance was AUC–ROC = 0.86, CI 0.81–0.88.

##### eICU Dataset

The RF classifier (training 1000 trees with deviance splitting criterion, minimum leaf size = 5) was chosen as the winning algorithm in the AutoML approach, using the extended feature approach. Among the various feature groups, the model with the best performance was trained with the dataset containing *all features* (AUC–ROC = 0.87, CI 0.84–0.9) and *labs* (AUC–ROC = 0.87, CI 0.83–0.9), followed by *drugs* (AUC–ROC = 0.82, CI 0.77–0.86) *vital signs* (AUC–ROC = 0.81, CI 0.76–0.85), whereas *medical history* (AUC–ROC = 0.6, CI 0.56–0.64) and *medications prior to admission* (AUC–ROC = 0.55, CI 0.5–0.59) had the worst performance, as shown in Figure 3. The AUC of the precision–recall curve was 0.45 regarding early mortality in ICU patients with VTE from the eICU dataset. The use of the extended feature set significantly outperformed traditional clinical scores APSIII and APACHE IVa. A detailed comparison of the various metrics of performance between the various feature groups and clinical scores is shown in Table 2. From an initial number of 2300 attributes in *all features* and 891 in the *labs* subsets, the Test-Budgeted Statistically Equivalent Signature (SES) algorithm (hyper-parameters: maxK = 2, alpha = 0.05, and budget = 3 · nvars) used a signature of only 25 features, predictive of early mortality, as shown in Figure 4a,b, respectively.

#### 5.1.2. Prediction of Early Mortality in ICU Patients with Cancer

We extracted 902 non-survivors vs. 1757 surviving patients with various solid cancer and hematological malignancies from the MIMIC-III database. SVM of type C-SVC with linear kernel (cost = 1.0) was the best-performing model for early mortality prediction (AUC–ROC = 0.94, CI 0.92–0.96). The Lasso feature selection [45] (penalty = 1.0, lambda = 4.183 × 10^−2^) algorithm revealed the following features with high predictive performance: endotrachial intubation, 1st day respiratory rate, coexistence of metastatic cancer, albumin, systolic blood pressure, and Red Cell Distribution Width (RDW) (Appendix A
Table A2). Using the all features set outperformed traditional medical scores OASIS (AUC–ROC = 0.83, CI 0.79–0.87), SAPS (AUC–ROC = 0.86, CI 0.82–0.9), and SOFA (AUC–ROC = 0.78, CI 0.73–0.83), as shown in Figure 5 and Table 3.

#### 5.1.3. Prediction of Late Mortality in ICU Patients with Thrombosis

Late mortality is defined as mortality after ICU or hospital discharge, as recorded in a later admission or outside the hospital. The MIMIC-III database is suitable for late mortality studies, since it offers longitudinal follow-up information regarding survival, for months after their admission to the ICU, in contrast to the eICU database, where such information is missing and it is not possible to chronologically order hospital admissions for the same patient within the same year. Moreover, no censor outcomes are recorded, since each admission has specific time stamps. For this binary classification task, 817 non-survivors vs. 1303 survivor patients with VTE from MIMIC-III were included. On average, patients with VTE died 549 days after admission, with a median of 225 days.

The best ML model was RF training 500 trees with the deviance splitting criterion and minimum leaf size = 3. As expected, the task of predicting late mortality was less efficient than that of the early mortality even using the whole feature space (AUC–ROC = 0.82, CI 0.79–0.84 vs. 0.93, CI 0.91–0.95). Although traditional clinical scores were originally designed to predict in-hospital mortality, their performance in predicting late mortality is moderate and acceptable. The SAPSII (AUC–ROC = 0.76, CI 0.72–0.81) and Elixhauser comorbidity scores (AUC–ROC = 0.74, CI 0.69–0.79) had the highest predictive performance from the rest of the studied scores and were close to the use of a combination of the available scores as shown in Figure 6.

Detailed performance metrics for both tasks of predicting early and late mortality using all features in patients with thrombosis, and from both datasets, are summarized in Table 4.

#### 5.1.4. Prediction of Late Mortality in ICU Patients with Cancer

Prediction of late mortality for cancer patients was further stratified based on the time of death after admission (in months). When constructing the predictive models based on time (months) after admission, AUC–ROC for months $m_{1}$, $m_{3}$, $m_{6}$, $m_{12}$ and $m_{>12}$ were 0.88, 0.84, 0.78, 0.76, 0.74, respectively.

Detailed performance metrics are shown in Table 3 and Figure 7. The best performance in the prediction of late mortality was achieved using *all features*, which outperformed classic clinical scores, especially in terms of F1 score, specificity, and sensitivity. Among the clinical scores, SAPS showed modest performance to predict mortality at 1 month after admission (AUC–ROC = 0.82, CI 0.78–0.85) and at 3 months after admission (AUC–ROC = 0.77, CI 0.73–0.81), although significantly lower than using the *all features* set. Features that were selected to have high predictive performance can be found in Appendix A
Table A2. The presence of metastatic cancer was the strongest predictor of mortality, whereas, interestingly, transfusions with red blood cells were the strongest predictor of mortality for more than a year after ICU admission.

#### 5.1.5. Model Validation

The predictive model for early mortality in patients with thrombosis derived from the MIMIC-III database ended with a signature of 35 predictive features that included features from full blood cell count (white blood cells (WBC), eosinophils, lymphocytes, and red cell distribution width (RDW)), biochemistry markers (glucose, blood urea nitrogen (BUN), calcium, total protein, albumin, potassium, lactate, lactate dehydrogenase (LDH), creatine phosphokinase (CPK), anion gap, arterial oxygen saturation (SaO2), pH), vital signs (heart rate, respiration, lowest systolic blood pressure (BP), and current diastolic BP), hemostasis (prothrombin time (PT), international normalised ratio (INR)), clinical information (cancer, sepsis, Glasgow Coma Scale (GCS)), and medications (vasopressors such as epinephrine, dopamine, and vasopressin and warfarin use). This model has been later validated in the eICU dataset, as shown in Table 5 and Figure 8.

### 5.2. Custom Machine Learning Framework: XGBoost

The best XGBoost classifier configuration (after hyper-parameter tuning) is with learning rate: 0.15, maximum tree depth: 12 gamma: 0.1, minimum sum of instance weight: 1, colsample_bytree: 0.7. Similarly to the AutoML framework, among the various feature groups, the model that achieved the best performance was the one trained with the dataset containing *all features* (AUC–ROC = 0.84, CI 0.83–0.85), as shown in Figure 9. The precision–recall curve AUC was 0.3 regarding early mortality in ICU patients with VTE from the eICU dataset, and this low score can be attributed to the extremely high imbalance class ratio. The detailed metrics of performance are shown in Table 6.

## 6. Discussion

The main contribution of the current work is the investigation of an extended number of clinico-laboratory features normally stored in the electronic health record, grouped into clinically meaningful sets (e.g., vital signs, labs, medications, and procedures) and in time stamps to examine their impact on the prediction of early and late mortality in ICU patients. Two homogeneous population cohorts based on their diagnosis (specifically venous thromboembolism and/or cancer) were derived using two different open-access large healthcare datasets, MIMIC-III and eICU. Patients with a DNR code were excluded as it is known that DNR is an independent factor of dismal survival outcomes in the ICU [49]. Specifically, we compared different state-of-the-art ML algorithms, addressed the class imbalance problem in the medical datasets, developed interpretable models and identified clinically meaningful predictive signatures, compared the performance of the ML approach with existing scoring systems, and finally provided an external validation of the model, proving the generalizability of it. Similarly to Choi et al. [10], our model outperformed traditional clinical scores not only in the prediction of early mortality but also in the prediction of late mortality. A multi-dimensional time series data-driven research approach was used, as well as stratification over the different groups of features, to identify feature groups with the highest predictive performance.

Towards constructing a robust model, training of two different ML strategies has been employed, an AutoML and a custom approach based on the XGBoost algorithm [50]. The primary aim of using the two approaches was not to compare them, since this it would be unfair, but to address the class imbalance through an oversampling method such as SMOTE [47]. Typically, class-imbalanced datasets constitute a common problem in medical informatics, which might lead to degraded performance depending on the type/number of data, features, etc., and thus an additional analysis should be performed in order to tackle this issue. JADBio addresses imbalanced classes through stratified cross-validation and diversified class weights during SVM learning. We thought that adopting the SMOTE method, which is considered a typical class balancing algorithm within the oversampling techniques framework [51], would be of added value. In the custom approach, we observed that the predictive accuracy was improved when features with more than XX missing values were removed. However, the accuracy appeared to remain constant despite the adoption of the SMOTE oversampling technique. Since SMOTE is a deterministic resampling method that selects examples being close in the local feature space, a probabilistic approach, such as Generative Adversarial Networks [52], which originally have been used in the area of image processing to produce synthetic but realistic images, could be used to produce new samples within a medical data analysis framework by learning from the overall class distribution. Moreover, the experimentation and extraction of the best-performing ML model in the custom approach is time-consuming since it requires substantial human and computational effort, artificial intelligence expertise, and extensive tuning of hyper-parameters; for this reason, automated ML tools are becoming popular among non-specialists in this area.

The AutoML approach based on the JADBio platform has been widely tested in biomedical data and follows all good practices for analysis and efficiency reporting [42]. During the experiments, the RF model was consistently found to be the winning algorithm, with a few exceptions. A frequent problem in various studies [53] that use ML predictive models is whether proper ML guidelines for over-fitting prevention and accurate performance metrics are reported. It is obvious from the experimental evaluation that JADBio can handle more efficiently than the custom ML framework high-dimensional datasets having a high level of missing values. Prediction of early mortality in patients with thrombosis was less efficient in the highly imbalanced dataset from the eICU compared with the one from the MIMIC-III dataset. The difference in performance, as illustrated in the precision–recall curve when compared to the ROC curve, is attributed to the highly imbalanced dataset. In this sense, for the minority class, high recall can only come at the cost of low precision. Feature discriminative analysis revealed that follow-up laboratory tests and vital signs in the medical record have the higher predictive performance when building prognostic models for early as well as late mortality, outperforming traditional clinical scores. Among traditional scores, the best performance for the prediction of early mortality in patients with thrombosis or cancer was demonstrated for SAPS [17]. Regarding late mortality in patients with thrombosis, SAPS and comorbidity showed modest performance, whereas in cancer patients, SAPS could modestly predict mortality up to 3 months after ICU admission.

The interpretability of the ML predictive models is considered a prerequisite for physicians in order to accept them as clinical decision support systems. To this end, both of our approaches produced interpretable and comparable models. The automated ML approach can produce “biosignatures” that can be intuitively explored and explained by physicians [43], as confirmed by this study. For feature selection, Statistically Equivalent Signature (SES) algorithms, inspired by the principles of constrained-based learning of Bayesian networks, were consistently found to be superior to the feature selection method Lasso. XGBoost has a built-in function for feature importance but different metrics can be used, which could lead to misleading results [54]. Although both approaches derived similar features, the ranking of importance was different.

Among the thousands of features that we extracted from the electronic health records of the patients with thrombosis, a few features were selected that were clinically meaningful, such as older age, cancer, respiratory, cardiovascular, and renal disease, vasopressor support, and mechanical ventilation, which are well established clinical predictors of ICU mortality [13]. Similarly to [13], sex was not found to be a predictor of ICU mortality. Moreover, individual feature analysis confirmed that warfarin [55], RDW [56], red blood cell transfusions [57], and blood urea nitrogen [58] are significant predictors of early and possibly long-term mortality. RDW has been shown to play a significant negative predictive role in ICU early mortality through the deregulation of erythropoiesis from inflammatory cytokines and oxidative stress [59]. It has also been reported to be an independent risk factor for cardiovascular diseases, dyslipidemia, diabetes, and renal and liver diseases. Surprisingly, high RDW has been shown to correlate with cancer stage irrespective of comorbidities, and with early mortality in VTE patients. For all these reasons, it is not paradoxical that RDW could be an easily applicable, new biomarker, useful not only for the prediction of early but possibly of late mortality in patients with thrombosis and/or cancer. Another interesting feature revealed by our prognostic model is eosinophil count, which is known to have prognostic significance in ICU patients [60]. Markers such as RDW and eosinophils are attractive since they are easily available and of low cost.

In patients with cancer, the strongest predictor of early and late ICU mortality is the presence of metastatic cancer. Red blood cell transfusions are a negative predictor of early mortality, as already known [61]. Interestingly, transfusions were found to be the strongest predictor of late mortality, more than one year after admission to the ICU, for patients with cancer. Red cell transfusions in patients with cancer not only increase the risk of death but also the risk of relapse [61]. Unfortunately, information regarding transfusion (red blood cells, plasma, and platelets) is missing in a significant number of patients, whereas in MIMIC-III, this information is scattered across two different information systems that collect data (Metavision and Carevue), and again, a significant number of data points are missing.

Head to head comparisons of the various studies in ICU mortality prediction are difficult, since the various studies have different inclusion and exclusion criteria, different types of studied features, and various definitions of mortality. Our study targeted two specific groups of patients, patients with venous thromboembolism and patients with cancer. Both diagnostic groups are high-risk patients, with a substantial risk of ICU admission and mortality. Mortality prediction models for ICU patients with thrombosis or cancer that are based on ML algorithms and use a large amount of clinical and laboratory data, structured and unstructured, are almost completely absent in the literature. Moreover, traditional scoring systems are not specific for these two diseases. To the best of our knowledge, only one publication on a relatively small number of patients with venous thromboembolism has been published [32]. Similarly to Runnan et al., we compared state-of-the-art ML algorithms with traditional scores, and we achieved comparable performance and identified similar predictive features.

In medical machine learning/predictive models, external validation in different datasets is imperative before clinical application. Classifiers usually perform well in the original dataset from which they were trained but then perform poorly in independent datasets. Another important consideration is that ML models in clinical practice are to be deployed in different institutions or countries. A recent systematic review identified only 5 out of 70 publications that used independent data to externally validate their model to predict mortality [53]. One of the strengths of our study is that it used two independent datasets to externally validate the ML prognostic model of early mortality derived from the MIMIC-III dataset to the eICU dataset, which are derived from different institutions and have different time periods. The results of the external validation were promising, with the exception of the F1 score, which was inferior.

Some limitations of this study should be considered. First of all, the study was based on retrospective data. Since the data were collected in the past, it is possible that many medical practices have changed over time, such as the case of warfarin use in the ICU. Second, the selection of the studied diagnostic groups was based solely on ICD-9 codes [62] and DRG codes [63], and not on imaging studies. Third, time series data were processed in specific time stamps, which increased significantly the dimensionality of the data. Moreover, we observed that labs and vital signs in both datasets were infrequently reported in the first 48 h, thus leading to a dramatically increased number of missing data on the various time stamps. Fourth, a direct comparison of our model with the only PE-specific score, PESI, was not possible, since this is not included in the datasets.

One of the primary goals of our future work is to directly compare our model with the PESI score and in a prospective cohort study. Inclusion of more features, such as genetic information and imaging studies, would be ideal and would probably improve the predictive performance. We could also focus on features extracted on the day of discharge to predict other outcomes, such as ICU readmission. Our future vision is to develop an intelligent ML-based system that is continuously updated with new clinical events and detailed information of the current clinical status of the patient, which could be a useful assistant for the physician and their clinical decision-making. To this end, the use of deep learning models, such as long short-term memory (LSTM) [14] for importing time series data in high-frequency datasets, and neural networks [64] could probably achieve better generalization performance with a significantly lower error rate. Shapley additive explanation (SHAP) analysis could be used to explain the output of our predictive model [65]. Handling of the high imbalance ratio of the datasets could be performed with other advanced resampling methods, such as Generative Adversarial Networks [52].

## 7. Conclusions

The presented research could be used as a proof of concept study that could be further validated in prospective or more recent datasets. Prediction of in-hospital mortality in patients with thrombosis or cancer is highly feasible, whereas prediction of late mortality is a more difficult and complex task. The results of this study are promising and, most importantly, interpretable, since the predictive features included in the model were clinically meaningful. The discovery of novel biomarkers, such as RDW and eosinophils, and their incorporation into the traditional clinical scores could possibly refine their performance.

## Figures and Tables

**Figure 1 ijms-23-07132-f001:**
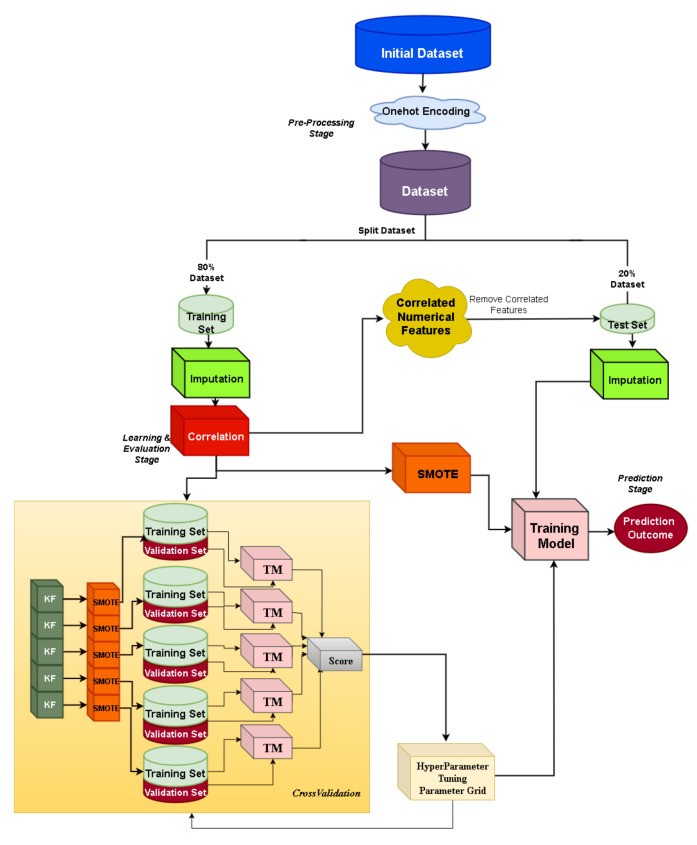
Custom machine learning pipeline: XGBoost.

**Figure 2 ijms-23-07132-f002:**
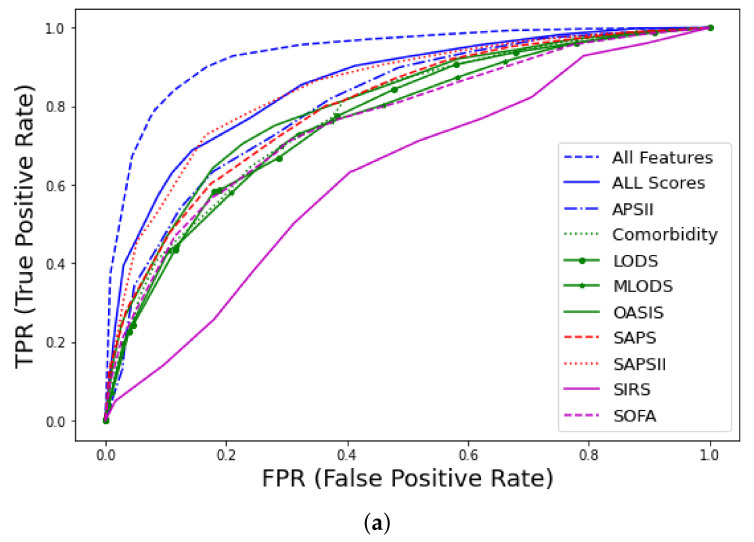
Receiver operating characteristic (ROC) curves for early mortality in ICU patients with thrombosis from MIMIC-III database using all features: (**a**) AUC–ROC curves and comparison with traditional scores (APS, Acute Physiology Score; LODS, Logistic Organ Dysfunction Score; MLODS, Multiple Logistic Organ Dysfunction Score; OASIS, Outcome and Assessment Information Set; SAPS, Simplified Acute Physiology Score; SIRS, Systemic Inflammatory Response Syndrome; SOFA, Sequential Organ Failure), (**b**) Precision–recall (PR) curve.

**Figure 3 ijms-23-07132-f003:**
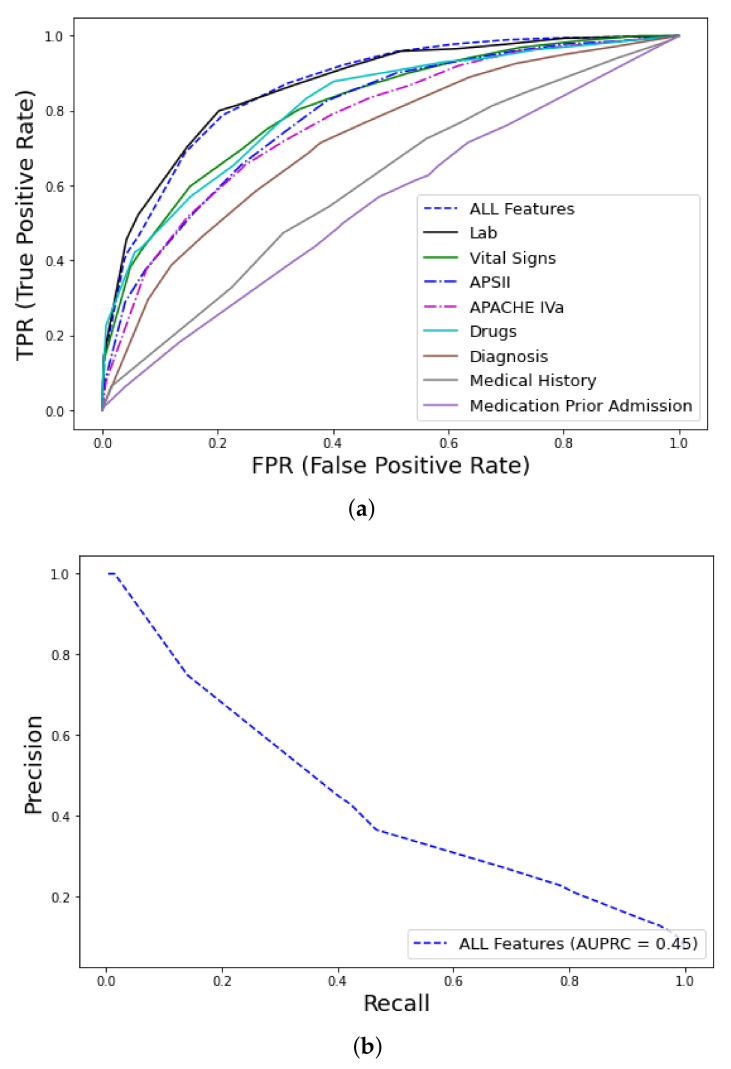
ROC curves for early mortality in ICU patients with VTE from eICU dataset: (**a**) AUC–ROC curves, feature discriminative analysis and comparison with clinical scores (APS, Acute Physiology Score; APACHE, Acute Physiology Age Chronic Health Evaluation), (**b**) Precision–recall curve.

**Figure 4 ijms-23-07132-f004:**
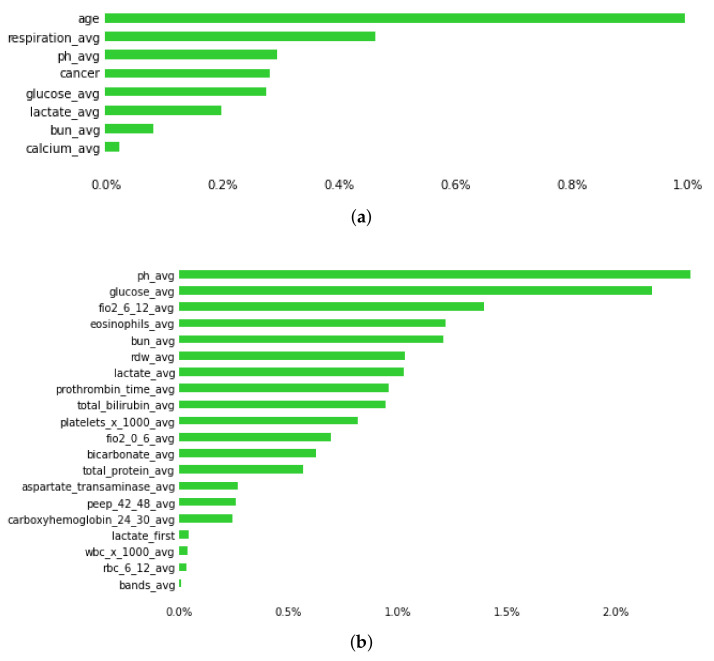
Plot showing predictive features of early mortality in ICU patients with thrombosis, using (**a**) allfeatures and (**b**) labs datasets. Green bars represent the average percentage drop in predictive performance when the feature is removed from the model. (Abbreviations: avg, average; bun, blood urea nitrogen; fiO2: fraction of inspired oxygen; rdw, red cell distribution width; peep, positive end expiratory pressure; wbc, white blood cells; 0–6, 6–12, 24–30, 42–48 is the time in hours after admission).

**Figure 5 ijms-23-07132-f005:**
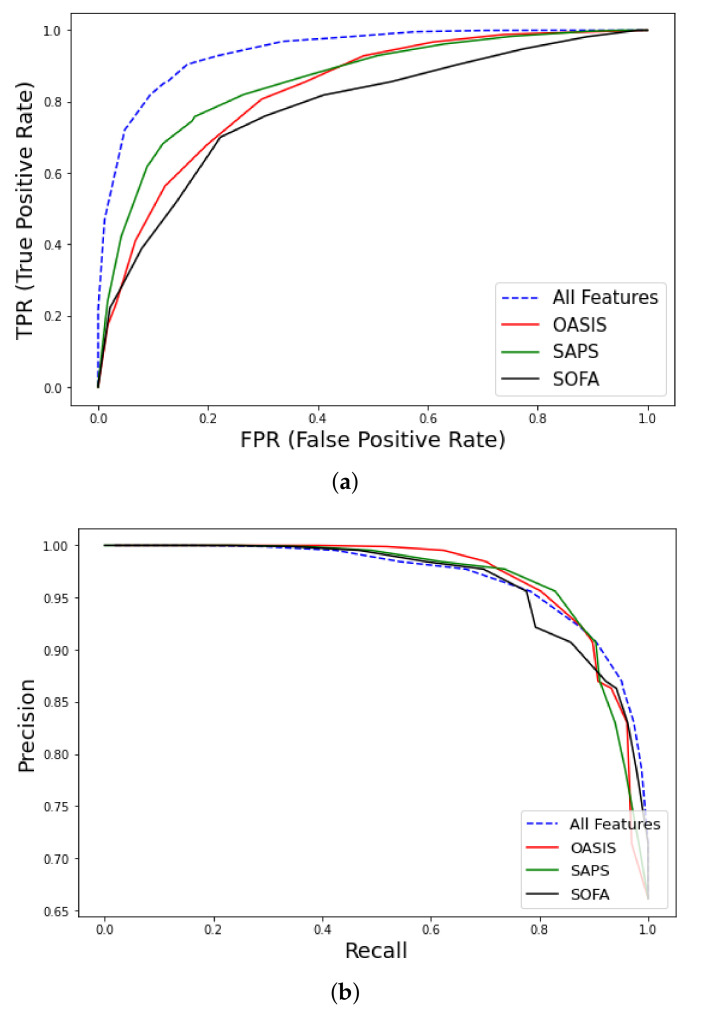
ROC curves for early mortality in ICU patients with cancer from MIMIC-III database using all features. (**a**) AUC–ROC curve and comparison with clinical scores (OASIS, Outcome and Assessment Information Set; SAPS, Simplified Acute Physiology Score; SOFA, Sequential Organ Failure), (**b**) Precision-Recall curve.

**Figure 6 ijms-23-07132-f006:**
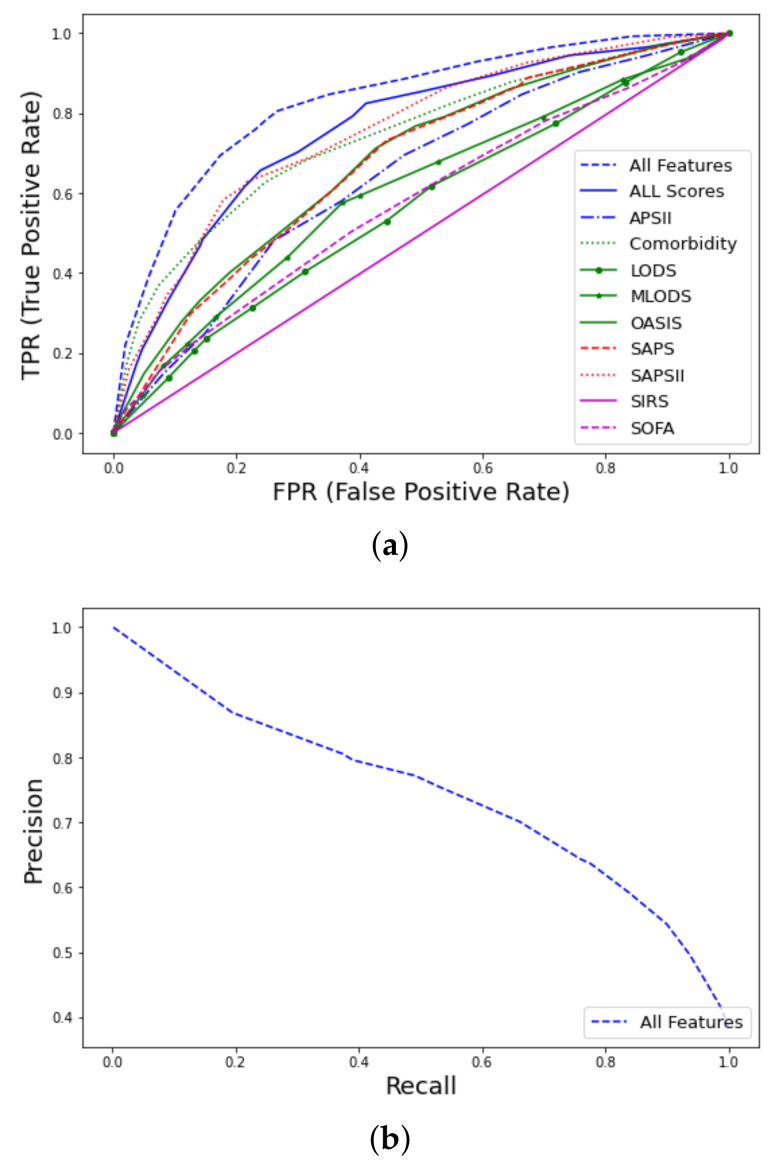
ROC curves for late mortality in ICU patients with thrombosis from MIMIC-III database using all features: (**a**) AUC–ROC curves and comparison with clinical scores. (**b**) PR curve.

**Figure 7 ijms-23-07132-f007:**
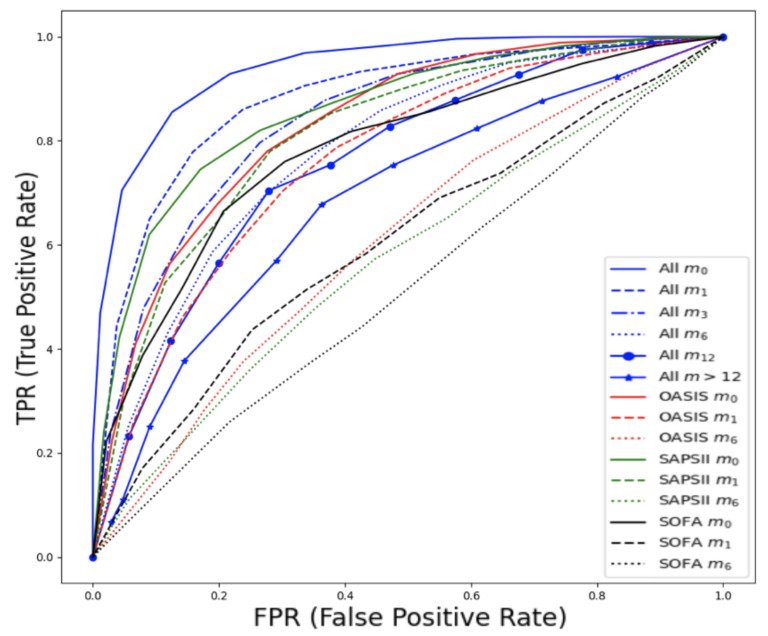
ROC curves for late mortality in ICU patients with cancer from MIMIC-III database, stratified based on the time of death after admission expressed in months ($m_{0}$, $m_{1}$, $m_{3}$, $m_{6}$, $m_{12}$, $>m_{12}$). Comparison with traditional clinical scores.

**Figure 8 ijms-23-07132-f008:**
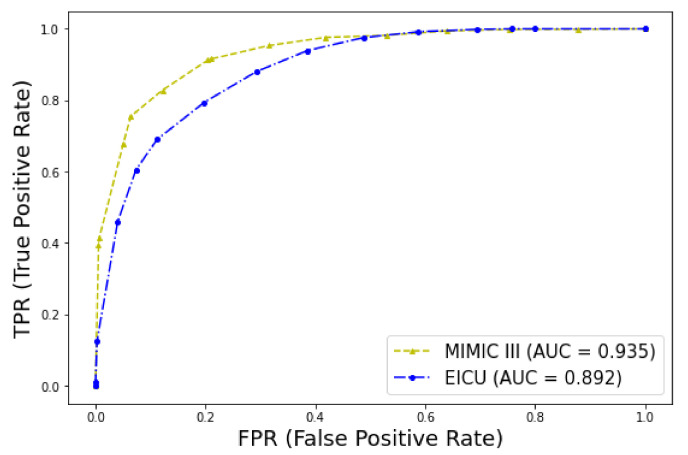
ROC curves for early mortality in ICU patients with thrombosis. Validation of the model in the two datasets.

**Figure 9 ijms-23-07132-f009:**
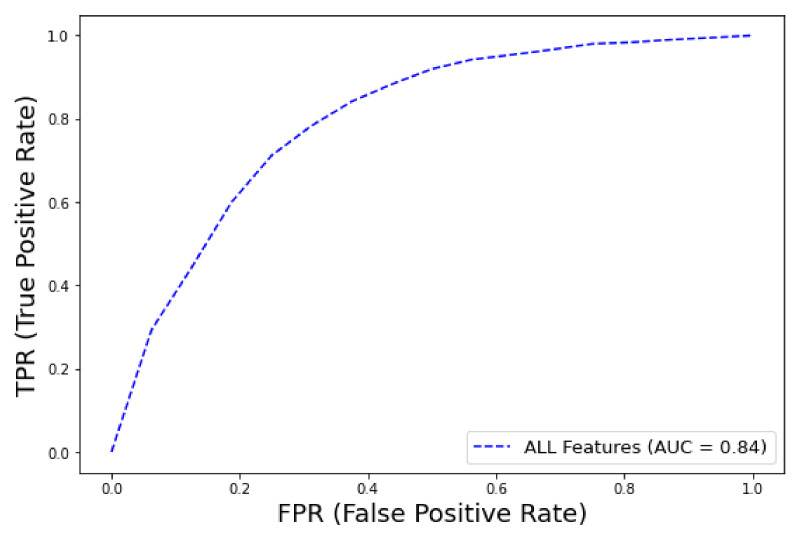
ROC curve for early mortality in ICU patients with VTE from eICU dataset (XGBoost).

**Table 1 ijms-23-07132-t001:** Demographic and clinical characteristics of patients with VTE or cancer from eICU and MIMIC-III databases. SD denotes the standard deviation, and LOS length of stay. *p*-Values between “surviving” and “non-surviving” patients are reported.

	VTE	VTE	Cancer			
Characteristic	eICU	*p*-Value	MIMIC-III	*p*-Value	**MIMIC-III**	*p*-Value
**Thrombosis** - **PE** - **DVT**	3724 2739 (62.4%) 2220 (50.6%)	0.71 0.95	2468 960 (38.9%) 1543 (62.5%)	0.92 -	5128 - -	
**Sex** - **Female** - **Male** - **Unknown**	1788 (48 %) 1934 (52 %) 1	0.24 0.24	1024 (41.5%) 1444 (58.5%) -	0.53 0.53	2192 (41.2%) 3126 (58.8%) -	0.07 0.07
**Ethnicity** - **White** - **Black** - **Other**	2845 (76.4%) 522 (14%) 357 (9.6%)	0.36 0.16	1801 (73%) 246 (10%) 421 (17%)	0.27 0.47	4049 (76%) 378 (7.1 %) 891 (16.8%)	0.57 0.83
**Age, years** **Average (SD)** **Min** **Max**	60.28 (16.26) 15 90	<0.001	62.64 (16.7) 17.4 98.7	<0.001	66.2 (14.2) 18.9 98.9	<0.001
**Cancer diagnosis**	409 (9.3 %)	<0.001	605 (24.5%)	<0.001		
**LOS, days** **Mean (SD)** **Median:**	11.22 (12.24) 7.01	0.002	7.06 (10.06), 153.9 days	<0.001	4.4 (6.5) 2.2	<0.001
**Mortality** (%): **Early:** **Late:** **“Survivor”:**	267 (7.16 %) N/A 3457 (92.84%)		348 (14.1%) 817 (33.1%) 1303 (52.8%)		902 (17%) 2659 (49.9%) 1757 (33.1%)	
**Death time (days)** **Average (SD):** **Median:**	N/A		390 (647) 83		328.5 (536.55) 365	

**Table 2 ijms-23-07132-t002:** Detailed metrics of performance for the predictive models of early mortality in patients with thrombosis from the eICU dataset (JADBio).

	AUC–ROC	F1 Score	Accuracy	Specificity	Sensitivity
**All features**	0.87 (0.84, 0.9)	0.42	0.92	0.95	0.42
**Labs**	0.87 (0.83, 0.9)	0.43	0.92	0.95	0.50
**Vital Signs**	0.83 (0.78, 0.87)	0.37	0.92	0.95	0.38
**APSIII**	0.79 (0.75, 0.84)	0.32	0.92	0.96	0.29
**APACHE IVa**	0.78 (0.73, 0.82)	0.30	0.89	0.92	0.38
**Drugs**	0.82 (0.77, 0.86)	0.53	0.94	0.99	0.23
**Diagnosis**	0.73 (0.68, 0.77)	0.26	0.82	0.92	0.30
**Medical History**	0.60 (0.56, 0.64)	0.27	0.93	0.98	0.06
**Medication prior to admission**	0.55 (0.5, 0.59)	0.41	0.92	0.99	0.01

**Table 3 ijms-23-07132-t003:** Detailed metrics of performance for prediction of early and late mortality in ICU patients with cancer using all features.

Mortality Prediction	AUC–ROC	Accuracy	F1 Score	Specificity	Sensitivity
**m0**	0.94 (0.92, 0.96)	0.88	0.82	0.91	0.82
**m1**	0.88 (0.85, 0.91)	0.85	0.89	0.84	0.78
**m3**	0.84 (0.8, 0.87)	0.79	0.85	0.74	0.79
**m6**	0.78 (0.74, 0.82)	0.72	0.49	0.72	0.71
**m12**	0.76 (0.71, 0.8)	0.72	0.5	0.72	0.7
**m > 12**	0.74 (0.69, 0.74)	0.68	0.76	0.64	0.74

**Table 4 ijms-23-07132-t004:** Detailed metrics of performance for prediction of early and late mortality of ICU patients with thrombosis using all features.

	Early Mortality		Late Mortality
	**MIMIC-III**	**eICU**	**MIMIC-III**
AUC–ROC	0.93	0.87	0.82
**Accuracy**	0.89	0.92	0.76
**F1 score**	0.72	0.97	0.60
**Sensitivity**	0.67	0.99	0.49
**Specificity**	0.95	0.1	0.90

**Table 5 ijms-23-07132-t005:** External validation of a predictive model for early mortality in patients with VTE based on a signature of 35 features.

	AUC–ROC	F1 Score	Accuracy	Specificity	Sensitivity
**MIMIC-III**	0.93 [0.91, 0.96]	0.74	0.77	0.93	0.74
**eICU**	0.89 [0.87, 0.91]	0.56	0.86	0.89	0.7

**Table 6 ijms-23-07132-t006:** Detailed metrics of performance for prediction of early mortality of ICU patients with thrombosis using all features.

XGBoost	All Features
AUC–ROC	0.84 (0.83–0.85)
**Accuracy**	0.87
**F1 score**	0.63
**Sensitivity**	0.29
**Specificity**	0.95

## Data Availability

Data from MIMIC-III and eICU datasets are available upon request from Physionet (www.physionet.org, accessed on 19 April 2022). Researchers must first complete a course regarding HIPAA requirements and then sign a data use agreement for appropriate data usage. ICD-9 and ICD-10 codes used in the research to find patients with thrombosis or cancer, source codes for replicating the custom-ML pipeline, and links to JADBio experiments are available upon request.

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
