# Peer review of "Outcome Prediction in Critically-Ill Patients with Venous Thromboembolism and/or Cancer Using Machine Learning Algorithms: External Validation and Comparison with Scoring Systems"

_ijms, 2022, doi:10.3390/ijms23137132_

Round 1
Reviewer 1 Report
This work is not enough contribution and innovation. However, the problem statement and motivation could be stronger or more clearly highlighted.
1. The existing literature should be classified and systematically reviewed, instead of being independently introduced one-by-one.
2. The abstract is too general and not prepared objectively. It should briefly highlight the paper's novelty as what is the main problem, how has it been resolved and where the novelty lies?
3. For better readability, the authors may expand the abbreviations at every first occurrence.
4. The author should provide only relevant information related to this paper and reserve more space for the proposed framework.
5. However, the author should compare the proposed algorithm with other recent works or provide a discussion. Otherwise, it's hard for the reader to identify the novelty and contribution of this work.
6. The descriptions given in this proposed scheme are not sufficient that this manuscript only adopted a variety of existing methods to complete the experiment where there are no strong hypothesis and methodical theoretical arguments. Therefore, the reviewer considers that this paper needs more works.
The algorithm presented has not any novelty.
7. The related works section is very short and no benefits from it. I suggest increasing the number of studies and add a new discussion there to show the advantage. Following can be included in related work
a. LDA–GA–SVM: improved hepatocellular carcinoma prediction through dimensionality reduction and genetically optimized support vector machine.
b. An approach based on mutually informed neural networks to optimize the generalization capabilities of decision support systems developed for heart failure prediction.
c. Machine-learning-scheme to detect choroidal-neovascularization in retinal OCT image.
d. An adaptive hybrid differential evolution algorithm for continuous optimization and classification problems
8. The manuscript is not well organized. The introduction section must introduce the status and motivation of this work and summarize with a paragraph about this paper.
Reviewer 2 Report
In this study, the authors proposed a machine learning model to predict early and late mortality of ICU patients with venous thromboembolism (VTE) and/or cancer. The model is conducted on 2 public datasets from MIMIC and eICU. Although it reached a promising performance, some major points should be addressed as follows:
1. The must have external validation data to evaluate the performance of models.
2. The authors mentioned that they built "explainable machine learning models", but we actually did not see any explainable ML in the study. The models included the use of AutoML and XGBoost without further explanation. Also, some explainable ML methods to interpret the features such as SHAP, LIME analyses were not considered.
3. Patient characteristics should be shown with p-values.
4. Detailed optimal hyperparameters of each model should be shown to help replicate the study.
5. Uncertainties of model should be reported.
6. Statistical tests should be conducted when comparing the predictive performance among methods/models.
7. Since the study is conducted on public datasets, they should compare the predictive performance to previously published works on the same problem/data.
8. More explanations/discussions should be shown in Figs. 4-5 since we could not catch the ideas behind the green box and the single lines.
9. Measurement metrics (i.e., sensitivity, specificity, accuracy, ...) are well-known and have been used in previous biomedical studies such as PMID: 34915158, PMID: 34502160. Thus, the authors are suggested to refer to more works in this description to attract a broader readership.
10. Source codes should be provided for replicating the study.
11. Besides ROC curves, the authors should show PR curves also.
Round 2
Reviewer 1 Report
The paper is well revised and can be accepted.
Reviewer 2 Report
My previous comments have been addressed.